# Association between behavioral phenotypes and response to a physical activity intervention using gamification and social incentives: Secondary analysis of the STEP UP randomized clinical trial

**Xisui Shirley Chen** [1] *, **Sujatha Changolkar**[2], **Amol S. Navathe**[1,3], **Kristin A. Linn**[4], **Gregory Reh**[5], **Gregory Szwartz**[5], **David Steier**[5], **Sarah Godby**[5], **Mohan Balachandran**[2], **Joseph D. Harrison**[2], **Charles A. L. Rareshide**[2], **Mitesh S. Patel**[1,2,3]

1 Department of Medicine, University of Pennyslvania Perelman School of Medicine, Philadelphia, Pennsylvania, United States of America, 2 Penn Medicine Nudge Unit, University of Pennyslvania, Philadelphia, Pennsylvania, United States of America, 3 Crescenz Veterans Affairs Medical Center, Philadelphia, Pennsylvania, United States of America, 4 Department of Biostatistics, Epidemiology and Informatics, University of Pennyslvania Perelman School of Medicine, Philadelphia, Pennsylvania, United States of America, 5 Deloitte Consulting, Philadelphia, Pennsylvania, United States of America

* shirley.chen@pennmedicine.upenn.edu

## Abstract

Participants often vary in their response to behavioral interventions, but methods to identify groups of participants that are more likely to respond are lacking. In this secondary analysis of a randomized clinical trial, we used baseline characteristics to group participants into distinct behavioral phenotypes and evaluated differential responses to a physical activity intervention. Latent class analysis was used to segment participants based on baseline participant data including demographics, validated measures of psychosocial variables, and physical activity behavior. The trial included 602 adults from 40 U.S. states with body mass index ≥25 who were randomized to control or one of three gamification interventions (supportive, collaborative, or competitive) to increase physical activity. Daily step counts were monitored using a wearable device for a 24-week intervention with 12 weeks of follow-up. The model segmented participants into three classes named for key defining traits: Class 1, extroverted and motivated; Class 2, less active and less social; Class 3, less motivated and at-risk. Adjusted regression models were used to test for differences in intervention response relative to control within each behavioral phenotype. In Class 1, only participants in the competitive arm increased their mean daily steps during the intervention (adjusted difference, 945; 95% CI, 352–1537; *P* = .002), but it was not sustained during follow-up. In Class 2, participants in all three gamification arms significantly increased their mean daily steps compared to control during the intervention (supportive arm adjusted difference 1172; 95% CI, 363–1980; *P* = .005; collaborative arm adjusted difference 1119; 95% CI, 319– 1919; *P* = .006; competitive arm adjusted difference 1179; 95% CI, 400–1957; *P* = .003) and all three had sustained impact during follow-up. In Class 3, none of the interventions had a significant effect on physical activity. Three behavioral phenotypes were identified, each

**Data Availability Statement:** All relevant data are within the manuscript and will be included in its Supporting Information files.

**Funding:** This study was funded by the University of Pennsylvania Health System through the Penn Medicine Nudge Unit. The University of Pennsylvania co-authors conducted all analyses, and drafted and submitted the manuscript. The Deloitte co-authors had the opportunity to review and provide comments on the manuscript. The University of Pennsylvania co-authors had full authority on whether or not to incorporate those comments. The Penn Medicine Nudge Unit provided support in the form of salaries for authors SC and CALR for this study. The funders had no role in study design, data collection and analysis, decision to publish, or preparation of the manuscript. The specific roles of all authors are articulated in the 'author contributions' section.

**Competing interests:** Dr. Patel was supported by career development awards from the Department of Veterans Affairs HSR&D and the Doris Duke Charitable Foundation. Dr. Patel is founder of Catalyst Health, a technology and behavior change consulting firm that has received consulting income from Deloitte, not related to this project. The following authors are or were employees of Deloitte: Gregory Reh, Gregory Szwartz, David Steier, and Sarah Godby. This does not alter our adherence to PLOS ONE policies on sharing data and materials. We declare no patents, products in development, or marketed products related to this study at this time.

with a different response to the interventions. This approach could be used to better target behavioral interventions to participants that are more likely to respond to them.

## Introduction

Modifiable behavioral risk factors such as poor diet, low physical activity, and tobacco use account for 40% of premature mortality in the United States, highlighting the need for effective behavior change interventions [1, 2]. Many behavior modification strategies appear promising, but systematic reviews and meta-analyses of randomized control trials indicate significant heterogeneity in outcomes that is poorly understood [3–5]. In particular, little is known about which interventions are effective for population sub-groups. While demographic attributes have been correlated with health behaviors [6], they often do not fully explain differences in intervention effectiveness [7–9]. Similarly, comparisons within studies suggest that baseline participant traits such as age, sex, race/ethnicity and health status are not reliable predictors of participant response [10–12].

Health behavior research could be improved by including additional data that informs factors influencing behavior change in intervention trials. For example, personality traits and psychological constructs such as affective response and resilience may elucidate important differences in how individuals perceive and engage with behavioral interventions that sociodemographic and clinical variables do not [12–14]. Evidence suggests that higher order combinations of demographic, health status, and psychosocial variables better capture the complexity of behavior change compared to any single variable type alone [15]. An analytical approach similar to audience segmentation in marketing can be applied to identify patterns of sample characteristics that describe baseline variation among participants [16]. In healthcare, segmentation analyses have been used to identify patient subgroups to better understand observed heterogeneity in clinical outcomes and care utilization [17–19]. Among several methods for subgroup identification, latent class analysis (LCA) provides a rigorous yet intuitive statistical approach that can accommodate a large number of variables and test competing models of segmentation.

The Social Incentives to Encourage Physical Activity and Understand Predictors (STEP UP) trial was a randomized clinical trial conducted by members of our group to test the effectiveness of supportive, collaborative, and competitive social incentives within a gamification intervention to increase physical activity among overweight and obese adults from 40 U.S. states [20]. While competition was found to be most effective on average, it may not have been the best intervention for all participants. The objective of this study was to use data on demographic, behavioral and psychological characteristics collected in the STEP UP trial to identify behavioral phenotypes and evaluate differential response to the interventions.

## Methods

### Study design

This is a secondary analysis of a previously conducted randomized clinical trial (ClinicalTrials.gov identifier: NCT03311230). The clinical trial's design, protocol, and main trial results have been previously published [20, 21]. In this study, we performed LCA of data collected during the main trial to identify participant behavioral phenotypes. We then compare changes in physical activity for each of the intervention arms relative to control within each behavioral

phenotype. The University of Pennsylvania Institutional Review Board approved the study and participants provided online informed consent during the clinical trial.

Briefly, STEP UP trial was designed to test the effectiveness of supportive, competitive, and collaborative social incentives to increase physical activity in a gamification intervention that incorporated principles of behavioral economics. The supportive arm asked participants to identify a friend or family member who was encouraged to support the participant at the start of the study and received weekly reports on the participant's performance; the competitive arm placed participants into groups of 3 and used a weekly leaderboard email to foster competition; the collaborative arm placed participants into groups of 3 and a designated member was selected each day to represent the team through their step activity.

Participants were recruited from Deloitte Consulting employees in the United States with a self-reported body mass index (BMI) of 25 or greater with a final sample of 602 employees from 40 U.S. states. During enrollment, participants were asked to complete a sociodemographic survey and a series of validated instruments to assess personality [22], risk taking behavior [23], grit [24], social support [25], exercise self-efficacy [26], mood [27], self-reported health status [28], sleep quality [29], and dietary patterns [30]. The primary outcome was change in daily steps from baseline to the 24-week intervention period. Step counts were tracked using a wrist-worn wearable device (Withings Activite Steel), which was mailed to all participants along with a replacement battery. These devices have been demonstrated to be accurate for tracking step counts [31]; devices were connected to Way to Health, a technology platform at the University of Pennsylvania used to deliver behavioral interventions [32]. The trial was conducted from February 2018 to March 2019 with a 2-week run-in period, a 24-week intervention period, and a 12-week follow-up period.

## Variable selection

Variables were selected as indicators in the LCA model based on conceptual relevance to health behavior change and known correlations to physical activity [7, 33]. We hypothesized that latent classes identified using relevant sociodemographic, psychological and behavioral variables would have differential response to the interventions. In addition, we examined the distribution of variables in our sample to exclude those without sufficient variability in distribution as they were unlikely to assist in differentiating subgroups. LCA uses discrete categorical variables as inputs [34], so all continuous variables were transformed into categorical variables based on survey scores or sample distribution as described below.

We selected variables from the following domains: 1) Demographics, which included age (grouped into 18–34 years, 35–49 years, and ≥50 years), sex, and previous use of wearable devices (yes/no). 2) Health behaviors, which included baseline physical activity and sleep quality. Baseline daily step count and coefficient of variance in baseline steps were used as measures of baseline physical activity level and consistency, respectively. They were calculated from the 2-week run-in period and converted into sample-based tertiles. Sleep quality was measured using the Pittsburg Sleep Quality Index survey and dichotomized into good sleep quality (total score <5) and poor sleep quality (total score ≥ 5) based on established interpretations of the scale [29]. 3) Personality, which included the Big Five traits of extroversion, agreeableness, conscientiousness, neuroticism, and openness [22], and grit, a measure of perseverance and passion for long-term goals [24]. 4) Behavioral perceptions, which included self-judgements of exercise self-efficacy and risk-taking specific to health and social behavior. Exercise self-efficacy was measured using the "sticking to it" factor score from the Self-Efficacy for Exercise Behaviors survey [26]. The DOSPERT survey was used to assess the likelihood to engage in risky behaviors related to health/safety and social interactions [23]. 5) Social support, which

was measured using the overall score on the Medical Outcomes Study (MOS) Social Support survey and calculated as an average of subscores in the domains of emotional support, tangible support, affectionate support, and positive social interactions [25]. The overall MOS score, each of the Big Five personality traits, grit, and exercise self-efficacy were scored on a 5-point scale; variables on a 5-point scale were converted into lower (1–2.9), medium (3–3.9), and higher (4–5) levels of each trait. The DOSPERT survey uses a 7-point scale for each domain and was similarly converted into lower (1–2.9), medium (3–4.9), and higher (5–7) levels of risk-taking preference.

### Statistical analyses

We used Mplus (Version 8.2), a common software package used for LCA [35], to perform the LCA with the indicator variables detailed above. A series of models fitting two through seven classes to the baseline data were sequentially generated and compared in an iterative approach for model selection. In LCA, the best model fit is achieved by evaluating quantitative model fit indices and considering the overall interpretability of the model, which involves qualitative investigator assessments of model parsimony and class distinctiveness [36]. First, statistical model fit indices were assessed together to identify models with lower Akaike information criterion (AIC) and Bayesian information criterion (BIC) values and greater entropy values [37, 38]. To test if adding another class improved model fit, we used the parametric bootstrapped likelihood ratio test [39]. Additionally, we considered number and size of classes to maximize model interpretability and statistical power. Models with classes containing less than 5% of the total sample were eliminated.

Once the best fit model was determined, we used descriptive statistics to compare baseline variables and step count outcomes among the latent classes in R (Version 3.5.1; R Foundation for Statistical Computing). Among the indicator variables used in our LCA model, we sought to determine which variables most strongly differentiated each class from the overall study population. Thus, probability weights for each variable category were generated in Mplus and used to calculate the proportion of participants in each variable category in each class compared to the overall sample (S1 Table). We used a cutoff of 1.33 or a 33% difference from the overall sample to highlight the characteristics driving class distinction.

Similar to the STEP UP trial [20], we fit generalized mixed-effects models to compare each intervention arm to control within each latent class for the intervention and follow-up periods. Models were adjusted for baseline step count, calendar month fixed effects, and study arm. To estimate the difference in change in steps between study arms, we used a least squared means approach. We used a conservative Bonferroni adjustment for 3 comparisons using a $P$ value of <0.017 for statistical significance. Regression analyses were conducted in SAS (version 9.4).

## Results

Participants had a mean (SD) age of 39 (10) years and BMI of 30 (5), 71% were male, and 64% had previously used a wearable activity tracking device. Baseline survey response rates were 100% (602/602). The mean (SD) baseline daily step count was 6204 (2646) steps and about half of the sample (49%) met criteria for poor sleep quality. Based on model fit parameters, the 3-class and 6-class model performed similarly with the lowest BIC and AIC values, respectively (S2 Table). The 3-class model was selected after considering model interpretability. The smallest class in the 3-class model still contained 20% of the sample population and clear variable patterns were qualitatively observed between classes. As the more parsimonious model, the 3-class model also offered more power to evaluate for differences between arms. All baseline variables except for the coefficient of variance in baseline steps were significantly different between the three

classes (Table 1). The three classes are described in detail below. Big Five personality traits are also described relative to the sample mean consistent with trait interpretation in the psychology literature. Table 2 lists the key characteristics of each class using probability weighted variables.

## Class 1- more extroverted and more motivated

Class 1 was the largest subgroup containing 54% of the study population. They were older in age (mean, 40.3 years) and 69% (226/328) had previously used a wearable device. Out of the three classes, this class had the highest scores in grit (+0.5 SD above the sample mean), conscientiousness (+0.5 SD) and exercise self-efficacy (+0.43 SD), indicating a greater degree of motivation and self-control. Other defining characteristics were higher levels of extroversion (+0.25 SD) and openness (+0.17 SD).

## Class 2- less active and less social

Class 2 comprised 20% of the study population. This class was more heavily male (82%, 99/121) with a lower baseline daily step count (mean, 5434 steps) and the largest percentage of members who had never previously used a wearable device (49%, 59/121). In contrast to class 1, members of class 2 had the lowest scores for extroversion (-0.63 SD below the mean) and openness (-0.67 SD). They had lower exercise self-efficacy but moderate levels of grit and conscientiousness. In addition, this class was characterized by lower social support and less social risk-taking behavior.

## Class 3- less motivated and at-risk

Class 3 comprised 25% of the study population. This class was youngest in age (mean, 36.1 years) and had more females (36%, 55/153) in comparison to the other classes. Members of class 3 had

**Table 1. Baseline LCA variables in study population and by class.**

|  | Class 1 | Class 2 | Class 3 | P-value | Overall |
|---|---|---|---|---|---|
|  | n = 328 | n = 121 | n = 153 |  | n = 602 |
| Age | 40.3 (10.6) | 37.8 (7.8) | 36.1 (10.6) | < .001 | 38.72 (10.2) |
| Male, N (%) | 230 (70.1) | 99 (81.8) | 98 (64.1) | 0.005 | 427 (70.9) |
| Previously used a wearable, N (%) | 226 (68.9) | 62 (51.2) | 96 (62.8) | 0.002 | 384 (63.8) |
| Baseline mean daily steps | 6308 (2537) | 5434 (2013) | 6591 (3163) | 0.002 | 6204 (2646) |
| Coefficient of variance baseline steps | 2.0 (0.8) | 1.9 (0.8) | 2.0 (0.8) | 0.29 | 2.0 (0.8) |
| Extroversion (0–5, 5 = Most extroverted) | 3.6 (0.8) | 2.9 (0.7) | 3.1 (0.8) | < .001 | 3.4 (0.8) |
| Agreeableness (0–5, 5 = Most agreeable) | 4.2 (0.6) | 3.8 (0.4) | 3.2 (0.6) | < .001 | 4.0 (0.6) |
| Conscientiousness (0–5, 5 = Most conscientious) | 4.2 (0.5) | 3.8 (0.4) | 3.0 (0.6) | < .001 | 3.9 (0.6) |
| Neuroticism (0–5, 5 = Most neurotic) | 2.2 (0.6) | 2.7 (0.6) | 3.4 (0.7) | < .001 | 2.5 (0.8) |
| Openness (0–5, 5 = Most open) | 3.8 (0.6) | 3.3 (0.4) | 3.7 (0.6) | < .001 | 3.7 (0.6) |
| ESE sticking to it (0–5, 5 = More likely to stick to exercise routine) | 4.1 (0.7) | 3.5 (0.6) | 3.4 (0.7) | < .001 | 3.8 (0.7) |
| PSQI overall (0–15, ≥5 = Poor sleep quality) | 4.4 (2.9) | 4.7 (2.7) | 6.4 (3.1) | < .001 | 5.0 (3.0) |
| MOS SS overall (0 to 5, 5 = More support) | 4.3 (0.8) | 3.9 (0.8) | 3.9 (0.9) | < .001 | 4.1 (0.9) |
| DOSPERT health/safety (1 to 7, 7 = More likely to engage in risky behavior) | 2.5 (1.0) | 2.2 (0.8) | 3.3 (1.1) | < .001 | 2.6 (1.1) |
| DOSPERT social (1 to 7, 7 = More likely to engage in risky behavior) | 5.1 (1.0) | 4.0 (0.8) | 4.9 (0.9) | < .001 | 4.8 (1.0) |
| Grit (0–5, 5 = Extremely gritty) | 4.0 (0.4) | 3.6 (0.3) | 3.0 (0.5) | < .001 | 3.7 (0.6) |

Data are presented as mean (SD) of participants unless otherwise indicated.

Abbreviations: ESE, Exercise Self-Efficacy survey; PSQI, Pittsburgh Sleep Quality Index; MOS SS, Medical Outcomes Survey Social Support; DOSPERT, Domain-Specific Risk-Taking scale.

**Table 2. Key factors driving class segmentation.**

| Class 1 "Extroverted and motivated" | Class 2 "Less active and less social" | Class 3 "Less motivated and at-risk" |
|---|---|---|
| Higher grit | Medium grit | Lower grit |
| Higher exercise self-efficacy | Lower exercise self-efficacy | Lower exercise self-efficacy |
| Higher extroversion | Lower extroversion | Lower extroversion |
| Higher conscientiousness | Medium conscientiousness | Lower conscientiousness |
| Higher openness | Lower openness | Higher neuroticism |
| | Lower social risk-taking | Higher health/safety risk-taking |
| | Lower social support | Lower social support |
| | Less prior wearable use | Lower agreeableness |
| | Lower baseline steps | Lower sleep quality |
| | | Younger age |

the highest scores for neuroticism (+1.12 SD above the mean) and lowest scores for agreeableness (-1.33 SD) and conscientiousness (-1.5 SD). Similar to class 2, members of class 3 had lower exercise self-efficacy, social support and extroversion (-0.38 SD), but tended to report more health-related risk-taking behavior. Finally, this class was notable for having the worst sleep scores among the 3 classes and were 57% more likely to have poor sleep quality than the overall sample.

## Intervention response by class

In adjusted analyses, class 1 participants had a significant change in mean daily step counts during the intervention period for only the gamification with competition arm (adjusted differences relative to control, 945 steps; 95% CI, 352–1537; $P$ = .002) (Table 3). However, there was no significant change in mean daily steps relative to control for any of the gamification arms during follow-up.

**Table 3. Adjusted model of change in mean daily step count by intervention arm for class 1 (extroverted and motivated).**

| | Class 1 (n = 328) | | | |
|---|---|---|---|---|
| | Control n = 71 | Gamification with Support n = 81 | Gamification with Collaboration n = 86 | Gamification with Competition n = 90 |
| **Baseline** | | | | |
| Steps per day, mean (SD) | 6021 (2612) | 6747 (2710) | 5992 (2056) | 6439 (2698) |
| **Main intervention period** | | | | |
| Steps per day, mean (SD) | 6248 (2220) | 7175 (2476) | 6743 (2103) | 7491 (3028) |
| **Primary model** | | | | |
| Difference relative to control and adjusted for baseline (95% CI) | - | 472.1 (-72.2, 1016.4) | 514 (-5.2, 1033.1) | 944.8 (352.4, 1537.3) |
| $P$ value[1] | - | 0.09 | 0.05 | 0.002 |
| **Follow-up period** | | | | |
| Steps per day, mean (SD) | 6187 (2437) | 6584 (2566) | 5850 (1442) | 6718 (2792) |
| **Main adjusted model** | | | | |
| Difference relative to control and adjusted for baseline (95% CI) | - | -38.5 (-674.1, 597.2) | -328.6 (-842.8, 185.5) | 266.1 (-371.2, 903.4) |
| $P$ value[1] | - | 0.91 | 0.21 | 0.41 |

Abbreviations: SD, standard deviation; CI, confidence interval.

[1]Bonferroni adjusted P value <0.017 used to determine significance.

Class 2 participants in all 3 gamification arms had a significant change in mean daily steps relative to control during the intervention period that was sustained during the follow-up period ([Table 4]). For class 2 participants in the supportive arm, the change in mean daily steps relative to control was 1172 (95% CI, 363–1980; $P = 0.005$) during the intervention period and 1061 (95% CI, 343–1669; $P = 0.003$) during the follow-up period. For class 2 participants in the collaborative arm, the change in mean daily steps relative to control was 1119 (95% CI, 319–1919; $P = 0.006$) during the intervention period and 1044 (95% CI 413–1675; $P = 0.001$) during the follow-up period. For class 2 participants in the competitive arm, the change in mean daily steps relative to control was 1179 (95% CI, 400–1957; $P = 0.003$) during the intervention period and 941 (95% CI, 269–1614; $P = 0.006$) during the follow-up period.

Among participants in class 3, there was no significant change in mean daily steps relative to control for any of the gamification arms during the intervention or follow-up periods. ([Table 5]).

## Discussion

In this secondary analysis of a randomized clinical trial, we demonstrated that baseline demographic, behavioral and psychological characteristics identified groups of participants with behavioral phenotypes that had differential response to the physical activity interventions. Rather than examining predictors in isolation, we used latent class analysis to reveal groups defined by co-occurring traits and behavioral patterns. These variable groupings constitute behavioral phenotypes that could provide a framework for selecting target populations of behavioral interventions [40]. While the main trial analyses showed that competition was most effective overall, our segmentation reveals substantial variations in intervention response by behavioral phenotype.

Class 1 participants performed best in the competitive arm and accounted for over half of the study population. However, they did not maintain their physical activity during follow-up

**Table 4. Adjusted model of change in mean daily step count by intervention arm for class 2 (less active and less social).**

| | Class 2 (n = 121) | | | |
| --- | --- | --- | --- | --- |
| | Control<br>n = 33 | Gamification with<br>Support<br>n = 30 | Gamification with<br>Collaboration<br>n = 29 | Gamification with<br>Competition<br>n = 29 |
| **Baseline** | | | | |
| Steps per day, mean (SD) | 5588 (1844) | 5407 (1995) | 5098 (1964) | 5624 (2306) |
| **Main intervention period** | | | | |
| Steps per day, mean (SD) | 5488 (1425) | 6556 (2393) | 6333 (2290) | 6693 (2404) |
| **Primary model** | | | | |
| Difference relative to control and adjusted for baseline (95% CI) | - | 1171.7 (363.3, 1980.1) | 1119.4 (319.8, 1919.1) | 1178.5 (400.3, 1956.6) |
| $P$ value[1] | - | 0.005 | 0.006 | 0.003 |
| **Follow-up period** | | | | |
| Steps per day, mean (SD) | 5039 (1062) | 5974 (1998) | 5888 (1738) | 6024 (1973) |
| **Main adjusted model** | | | | |
| Difference relative to control and adjusted for baseline (95% CI) | - | 1006.1 (343, 1669.2) | 1043.7 (412.9, 1674.6) | 941.4 (268.7, 1614) |
| $P$ value[1] | - | 0.003 | 0.001 | 0.006 |

Abbreviations: SD, standard deviation; CI, confidence interval.

[1]Bonferroni adjusted P value <0.017 used to determine significance.

**Table 5. Adjusted model of change in mean daily step count by intervention arm for class 3 (less motivated and at-risk).**

| | Class 3 (n = 153) | | | |
| --- | --- | --- | --- | --- |
| | Control<br>n = 47 | Gamification with Support<br>n = 40 | Gamification with Collaboration<br>n = 35 | Gamification with Competition<br>n = 31 |
| **Baseline** | | | | |
| Steps per day, mean (SD) | 6533 (3073) | 6053 (2513) | 7282 (3628) | 6592 (3482) |
| **Main intervention period** | | | | |
| Steps per day, mean (SD) | 6505 (2366) | 6884 (2267) | 7387 (2286) | 7009 (3102) |
| **Primary model** | | | | |
| Difference relative to control and adjusted for baseline (95% CI) | - | 620.3 (-160, 1400.7) | 545.4 (-247.4, 1338.2) | 466.2 (-348.1, 1280.4) |
| P value[1] | - | 0.12 | 0.18 | 0.26 |
| **Follow-up period** | | | | |
| Steps per day, mean (SD) | 6068 (2063) | 6467 (2006) | 6624 (1908) | 6758 (2945) |
| **Main adjusted model** | | | | |
| Difference relative to control and adjusted for baseline (95% CI) | - | 593.2 (-152.4, 1338.8) | 273.9 (-472.2, 1019.9) | 662.6 (- 205.4, 1530.6) |
| P value[1] | - | 0.12 | 0.47 | 0.13 |

Abbreviations: SD, standard deviation; CI, confidence interval.

[1]Bonferroni adjusted P value <0.017 used to determine significance.

despite high levels of exercise self-efficacy, conscientiousness and grit. It may be that these traits enable individuals to perform favorably under social influence and comparison but have a smaller influence on habit formation. Sustaining behavior change is known to be challenging after incentives are withdrawn [41]. Ongoing incentive programs may be necessary to support many individuals, including those with high baseline motivation and self-control, and should be explored in future work.

Notably, class 2 participants significantly increased and sustained their physical activity in all three gamification arms through the follow-up period. Class 2 may represent the ideal participant phenotype for a remotely monitored gamification intervention that uses behavioral economics and social influence to promote physical activity. Since members of this class were less extroverted, less open, less likely to engage in social risk taking, and reported less social support, offering structured social incentives may be particularly impactful. Additionally, this group was less active and reported lower exercise self-efficacy at baseline, but had moderate levels of conscientiousness and grit, traits theorized to be protective for health [14]. The persistence of intervention effects in the follow-up period and similar outcomes seen across the intervention arms indicates that interventions may have facilitated habit formation for regular physical activity among individuals within this phenotype.

On the other hand, class 3 participants did not appear to benefit from any of the interventions. This group demonstrated traits seen in Type D personality or distressed personality type, a well-established phenotype characterized by negative affect and social inhibition which correlates with low extraversion, low conscientiousness and high neuroticism [42]. This personality type has been associated with a range of negative mental and physical health outcomes, felt to be mediated by poor health behaviors and maladaptive coping mechanisms [43]. Thus, the findings of worse sleep quality and more health-related risk-taking behaviors in this group is also consistent. Individuals with these characteristics could require more intensive cognitively designed interventions to support behavior change through mechanisms such as improving self-efficacy [44].

This is one of the first studies to use LCA to identify behavioral phenotypes and evaluate variations in participant response in a randomized clinical trial for increasing physical activity. Prior studies in this area have used recursive partitioning methods which uses cutpoints to split predictor variables [15, 45]. However, without careful application and parameter tuning, these flexible methods can be prone to overfitting, resulting in unstable decision trees and classification structures [46]. LCA represents an alternative approach to segmentation that can be evaluated by likelihood-based model fit indices with growing applications to healthcare in the era of personalized medicine. In the context of health behavior interventions, it is well-suited to synthesize the large number and wide range of potential predictors of success and the complex interactions between them that remain poorly understood and likely vary between populations. For example, LCA has been used to identify subgroups among African American and Chinese American patients with low rates of colon cancer screening based on sociodemographics and an expanded set of psychosocial risk factors including perceived barriers, health beliefs and cultural values [47, 48]. However, prospective studies that test targeted and behaviorally designed interventions are lacking.

Among digital health interventions for physical activity, there is evidence that leveraging social networks and incorporating features such as social support and social comparison in addition to self-monitoring can enhance engagement and retention [49]. At the same time, participants have expressed a range of preferences and concerns regarding these intervention features. In our study, class 2 participants increased their physical activity in all 3 intervention arms, suggesting that the type of social incentive used was not a major determinant of intervention response for this subgroup. Further studies are needed to explore differential response to social incentives in behavioral interventions, the role of stated participant preferences, and whether an ideal participant phenotype exists for specific incentives.

## Limitations

This study has several limitations. First, it is a secondary analysis of a randomized clinical trial. Our main objective was to use LCA to identify subgroups in the study population as a framework for understanding differential intervention effects [50]. The main trial was not designed to detect differences in step count between intervention and control among three latent classes so the analyses in the current study may be underpowered. Second, LCA only reveals patterns in variation present at baseline without consideration of the outcome. This approach does not formally test how to target identified subgroups or compare the relative benefits of targeting techniques, which will require alternative statistical prediction models and prospective studies. Third, this population of overweight and obese adults was from a large U.S. consulting firm, limiting the generalizability of our findings. Nonetheless, the distinct phenotypes present in this population suggest the importance of accounting for individual differences in personality, behavioral tendencies and social resources when designing and evaluating wellness programs.

## Conclusions

We identified three behavioral phenotypes among overweight and obese adult participants of a randomized clinical trial of a gamification intervention to increase physical activity. The phenotypes were characterized by a combination of data on demographics, health behaviors and psychological measures and were associated with different intervention outcomes. These findings demonstrated that a behavioral phenotyping approach can reveal differences in intervention response and identify those who are most likely to benefit. Future studies should establish and validate classifications of behavioral phenotypes in broader populations and formally test interventions targeted to specific participant phenotypes.

## Supporting information

**S1 Table. Variable weights by class.**
(DOCX)

**S2 Table. Fit statistics and number of individuals per class for latent class models for two to seven classes.**
(DOCX)

**S1 Dataset. Raw data used for STEP UP LCA analyses.**
(XLSX)

## Author Contributions

**Conceptualization:** Gregory Reh, Gregory Szwartz, David Steier, Mitesh S. Patel.

**Data curation:** Joseph D. Harrison.

**Formal analysis:** Xisui Shirley Chen, Sujatha Changolkar, Charles A. L. Rareshide.

**Funding acquisition:** Gregory Reh, Gregory Szwartz, David Steier, Mitesh S. Patel.

**Investigation:** Xisui Shirley Chen, Mitesh S. Patel.

**Methodology:** Xisui Shirley Chen, Sujatha Changolkar, Charles A. L. Rareshide, Mitesh S. Patel.

**Project administration:** Mohan Balachandran, Joseph D. Harrison, Mitesh S. Patel.

**Resources:** Mohan Balachandran, Mitesh S. Patel.

**Supervision:** Mitesh S. Patel.

**Writing – original draft:** Xisui Shirley Chen, Mitesh S. Patel.

**Writing – review & editing:** Xisui Shirley Chen, Sujatha Changolkar, Amol S. Navathe, Kristin A. Linn, Gregory Reh, Gregory Szwartz, David Steier, Sarah Godby, Mohan Balachandran, Joseph D. Harrison, Charles A. L. Rareshide, Mitesh S. Patel.

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
