## [Decision Letter · Decision Letter 0]

4 Jun 2020

PONE-D-20-11106

Association between Behavioral Phenotypes and Response to a Physical Activity Intervention Using Gamification and Social Incentives: Secondary Analysis of the STEP UP Randomized Clinical Trial

PLOS ONE

Dear Dr. Chen,

Thank you for submitting your manuscript to PLOS ONE. After careful consideration, we feel that it has merit but does not fully meet PLOS ONE’s publication criteria as it currently stands. Therefore, we invite you to submit a revised version of the manuscript that addresses the points raised during the review process.

We look forward to receiving your revised manuscript.

Kind regards,

Rebecca A Krukowski

Academic Editor

PLOS ONE

Journal Requirements:

1- Please ensure that your manuscript meets PLOS ONE's style requirements, including those for file naming. The PLOS ONE style templates can be found at https://journals.plos.org/plosone/s/file?id=wjVg/PLOSOne_formatting_sample_main_body.pdf and https://journals.plos.org/plosone/s/file?id=ba62/PLOSOne_formatting_sample_title_authors_affiliations.pdf

"This study was supported by Deloitte Consulting LLP and the University of Pennsylvania Health System through the Penn Medicine Nudge Unit. Funding was obtained by Mitesh Patel. The University of Pennsylvania conducted all analyses, and drafted and submitted the manuscript. The co-authors from the funding organization had the opportunity to review and provide comments on the manuscript. The University of Pennsylvania co-authors had full authority on whether or not to incorporate those comments. Deloitte was involved in study conceptualization but had no role in data collection and analysis, decision to publish, or preparation of the manuscript."

We note that one or more of the authors are employed by a commercial company: Deloitte Consulting.

2.2. Please provide an amended Funding Statement declaring this commercial affiliation, as well as a statement regarding the Role of Funders in your study. If the funding organization did not play a role in the study design, data collection and analysis, decision to publish, or preparation of the manuscript and only provided financial support in the form of authors' salaries and/or research materials, please review your statements relating to the author contributions, and ensure you have specifically and accurately indicated the role(s) that these authors had in your study. You can update author roles in the Author Contributions section of the online submission form.

2.2. Please also provide an updated Competing Interests Statement declaring this commercial affiliation along with any other relevant declarations relating to employment, consultancy, patents, products in development, or marketed products, etc. 

Reviewers' comments:

Reviewer's Responses to Questions

**Comments to the Author**

1. Is the manuscript technically sound, and do the data support the conclusions?

Reviewer #1: Yes

Reviewer #2: Yes

Reviewer #3: Yes

2. Has the statistical analysis been performed appropriately and rigorously? 

Reviewer #1: Yes

Reviewer #2: Yes

Reviewer #3: I Don't Know

3. Have the authors made all data underlying the findings in their manuscript fully available?

Reviewer #1: Yes

Reviewer #2: No

Reviewer #3: Yes

4. Is the manuscript presented in an intelligible fashion and written in standard English?

Reviewer #1: Yes

Reviewer #2: Yes

Reviewer #3: Yes

5. Review Comments to the Author

Reviewer #1: This manuscript presents a secondary analysis of data from a randomized clinical trial using latent class analysis to determine whether subgroups of participants responded different to the physical activity intervention. Overall, the methods and analytical approach are sound, and the results are interesting. The manuscript is well-written and succinct. I have a few recommendations for changes that I believe would further enhance the manuscript.

General comments:

1. I can appreciate that coming up with succinct, accurate names for each of the groups is a challenge. I thought the name for Class 2 was a good reflection of its defining characteristics. However, for Class 1 something that highlights their grit and conscientiousness might be more appropriate, as motivation was not directly assessed and extroversion was not as far above the mean as some of the other variables. For Class 3, again motivation was not assessed and the term “at-risk” is quite vague – at-risk for low physical activity? For poor mental health? I don’t have a suggestion for a perfect name for these groups, but I would encourage the authors to consider whether the current names can be improved.

2. I also appreciated that the authors did not overstate their findings and kept the discussion limited to this specific trial (e.g. p. 15, lines 294-6, “the ideal participant phenotype for a remotely monitored gamification intervention that uses behavioral economics and social influence to promote physical activity.”). On the other hand, this is a very specific intervention type, and readers may wonder whether these findings would be relevant to other types of behavioral interventions. Although there may be a limited number of studies that have used this exact approach (e.g., LCA in the context of physical activity RCTs), there are a variety of studies that have examined how subgroups of the total participant population respond differently to interventions. I would like to see the discussion incorporate some additional literature to put these results in the broader context of what we already know.

3. Similar to the comment above, I think it would be helpful to incorporate some additional discussion about how these types of studies can be used to enhance future interventions. The authors provide some general comments about targeting subgroups, but from a practical perspective, how might this be done? Based on this study and others like it, are there specific variables that should be consistently measured at baseline and used to determine what types of support participants may need? Are there evidence-based examples of different types of support being effective for subgroups of participants?

Specific comments:

p. 5, line 107: was the follow-up measure taken at the end of the 24-week intervention period? Were step counts averaged across a week at each time point?

p. 5, line 107-8: Did all participants wear the same wearable device? What device was it?

The manuscript sometimes uses “randomized clinical trial” and other times “randomized control trial” – be consistent throughout.

Reviewer #2: PONE-D-20-11106: statistical review

SUMMARY. This is a two-step secondary analysis of a previous clinical trial of daily changes of step counts (primary outcome). Subjects are first segmented according to three groups, using latent class methods. Then, generalized mixed-effects models are estimated to compare each intervention arm to control within each latent class for the intervention and follow-up periods. I have very little to say about this paper. Latent class analysis is correct and generalized linear mixed-effects models appropriately account for latent heterogeneity and longitudinal correlation of the outcomes. I just list below some minor points that the authors should address to improve clarity and reproducibility of the results.

MINOR POINTS

1. Table 2 is obtained by exploiting the conditional probabilities of the observed variables given the latent class: the table of these probabilities should be included in the paper, perhaps as supplementary material.

2. I guess that the mixed models that have been exploited are those of doi: 10.1001/jamainternmed.2019.3505. As a courtesy to the reader, please specify the kind of model you are using.

3. Please clarify that entropy (Table S1) is a measure of class separation and that, as such, it can be used for model selection.

4. Although data are available without restriction, they are not attached. Data should be inlcuded as supplementary material and metadata provided.

Reviewer #3: The aim of this secondary analysis of a remotely delivered randomized clinical physical activity promotion trial was to identify behavioral phenotypes and compare physical activity changes for each intervention arm relative to control within each behavioral phenotype. This study uniquely fills a gap in the physical activity promotion literature, helping advance our understanding of which gamification interventions – a promising and increasingly popular behavior change approach – may be most effective for different population subgroups. The identification and evaluation of behavioral phenotypes that are based on a combination of variables as opposed to a single variable is an insightful perspective rooted in a sound conceptual basis that has been successfully applied in other sectors. The large sample, consideration of variables relevant to physical activity that could be screened somewhat easily, and evaluation of physical activity change both at the end of the intervention and follow up are study strengths. The rationale for the analytic approach is well described, key limitations are highlighted, and reasonable interpretations of the findings are made. Some very minor concerns are listed below.

Minor Concerns

Introduction

• p. 3, line 60: Could you please clarify or provide a few examples in parentheses of what you mean by baseline participant traits to help differentiate from the variables you examined?

Methods

• Although it is mentioned in your main outcomes manuscript for the trial, could you please mention the wearable device that was used? Also, is it a wrist-worn device?

Discussion

• Your interpretations of your findings are logical. Nonetheless, are there any findings at all from previous physical activity promotion studies that could be discussed to help situate your findings in the broader literature – anything similar or different from your findings?

• It may not be a good idea to speculate too much, but do you have any thoughts about what could be explored in the future to help those who may have a more competitive orientation sustain their physical activity improvements? Or perhaps it could just be noted as a future area of inquiry if space permits.

• It may be worth mentioning that only daily step counts were examined – other outcomes may be of interest too and could be explored in future research (MVPA etc.).

6. PLOS authors have the option to publish the peer review history of their article (what does this mean?). If published, this will include your full peer review and any attached files.

Reviewer #1: No

Reviewer #2: No

Reviewer #3: No

---

## [Author Response · Author response to Decision Letter 0]

16 Jul 2020

We have included a point by point response to reviewer comments in this resubmission. The funding statement has also been revised to clarify the role of Deloitte and is included in our cover letter.

---

## [Decision Letter · Decision Letter 1]

3 Sep 2020

Association between Behavioral Phenotypes and Response to a Physical Activity Intervention Using Gamification and Social Incentives: Secondary Analysis of the STEP UP Randomized Clinical Trial

PONE-D-20-11106R1

Dear Dr. Chen,

We’re pleased to inform you that your manuscript has been judged scientifically suitable for publication and will be formally accepted for publication once it meets all outstanding technical requirements.

Kind regards,

Rebecca A Krukowski

Academic Editor

PLOS ONE

Additional Editor Comments (optional):

Reviewers' comments:

Reviewer's Responses to Questions

**Comments to the Author**

1. If the authors have adequately addressed your comments raised in a previous round of review and you feel that this manuscript is now acceptable for publication, you may indicate that here to bypass the “Comments to the Author” section, enter your conflict of interest statement in the “Confidential to Editor” section, and submit your "Accept" recommendation.

Reviewer #1: All comments have been addressed

Reviewer #2: All comments have been addressed

2. Is the manuscript technically sound, and do the data support the conclusions?

Reviewer #1: Yes

Reviewer #2: (No Response)

3. Has the statistical analysis been performed appropriately and rigorously? 

Reviewer #1: Yes

Reviewer #2: (No Response)

4. Have the authors made all data underlying the findings in their manuscript fully available?

Reviewer #1: Yes

Reviewer #2: (No Response)

5. Is the manuscript presented in an intelligible fashion and written in standard English?

Reviewer #1: Yes

Reviewer #2: (No Response)

6. Review Comments to the Author

Reviewer #1: (No Response)

Reviewer #2: (No Response)

7. PLOS authors have the option to publish the peer review history of their article (what does this mean?). If published, this will include your full peer review and any attached files.

Reviewer #1: No

Reviewer #2: No

---

## [Editor Report · Acceptance letter]

22 Sep 2020

PONE-D-20-11106R1 

Association between behavioral phenotypes and response to a
physical activity intervention using gamification and social incentives:
Secondary analysis of the STEP UP randomized clinical trial 

Dear Dr. Chen:

I'm pleased to inform you that your manuscript has been deemed suitable for publication in PLOS ONE. Congratulations! Your manuscript is now with our production department. 

Kind regards, 

on behalf of

Dr. Rebecca A Krukowski 

Academic Editor

PLOS ONE